# The Theoretical Framework of the Clinical Pilates Exercise Method in Managing Non-Specific Chronic Low Back Pain: A Narrative Review

**DOI:** 10.3390/biology10111096

**Published:** 2021-10-25

**Authors:** Boon Chong Kwok, Justin Xuan Li Lim, Pui Wah Kong

**Affiliations:** 1Physical Education and Sports Science Academic Group, National Institute of Education, Nanyang Technological University, Singapore 637616, Singapore; kwokboonchong@gmail.com; 2Rehabilitation, Clinical Pilates Family Physiotherapy, Singapore 079906, Singapore; justinlxl90@gmail.com

**Keywords:** exercise science, physiotherapy, rehabilitation, lumbago, motor control

## Abstract

**Simple Summary:**

Exercise is important in helping people with chronic low back pain to regain wellness. General exercises such as cycling and strength training helps with the condition but require a longer time to achieve meaningful improvements in pain and function. Movement preference is potentially useful in exercise and can help an individual to achieve improvements faster. The Clinical Pilates exercise method uses movement preference and thus is a hybrid of two of the best exercise techniques in managing chronic low back pain. However, current studies in Clinical Pilates are lacking and thus challenge the translation of the technique into clinical practice. Hence, a narrative review of the theory of the Clinical Pilates exercise method is examined, and current literature is reviewed to provide a guide towards successful exercise prescription. A structured approach to physical assessment of human movement is proposed to guide clinicians or researchers involved in exercise prescription to design effective exercises. The structured assessment approach also helps with managing clinical cases with multiple episodes of injuries. Despite limited evidence, the Clinical Pilates exercise method is safe and provides faster and earlier recovery and the same longer term outcomes as general exercises.

**Abstract:**

Exercise plays an important role in rehabilitating people with chronic low back pain. Aerobic exercise and resistance training are general exercise strategies to manage chronic low back pain, but these strategies require longer intervention period to achieve clinical outcomes in pain reduction and functional improvements. Directional preference is recognised as an important exercise strategy in managing low back pain. The Clinical Pilates exercise method leverages on the directional preference of an individual to achieve clinical outcomes faster. Clinical Pilates is a hybrid of two of the best exercise interventions for low back pain, which are general Pilates and the McKenzie method. Due to the scarcity of Clinical Pilates literature, a review of its theory and studies was undertaken to provide a structured guide to the technique in managing people with chronic low back pain. Hypothetical algorithms are developed to support translation into clinical practice and future research studies. These algorithms are useful in the management of complex cases involving multiple directional trauma. Although limited, current evidence suggests that the Clinical Pilates exercise method is safe and provides faster functional recovery in the early stage of rehabilitation and similar longer term outcomes as general exercises.

## 1. Introduction

Chronic low back pain (LBP) is a common musculoskeletal pain disorder that affects most adults and has the highest prevalence (65%) among other chronic musculoskeletal pain disorders [1]. Although spinal degeneration is a common aging process, about 90% of the LBP cases in primary care were not associated with a specific structural cause [2,3]. Thus, these cases were classified as non-specific chronic LBP [4]. Impairment to the motor control of the body could lead to non-specific LBP [5]. Hence, exercise continues to be a main conservative management strategy for non-specific chronic LBP [6,7]. Among the exercise strategies, there is an increasing trend in the use of Pilates exercises to manage non-specific chronic LBP [8,9,10]. However, a few variations of Pilates intervention exist in the management of non-specific chronic LBP [11,12,13], in which their approaches were not explored in systematic reviews [9,10]. The gap in exploring how the intervention approaches work was similarly identified in a recent review on motor control exercises for adults with non-specific LBP [14].

A research survey revealed that physiotherapists perceived direction-specific exercise as an important consideration in exercise prescription for people with non-specific chronic LBP [15]. However, a majority of fresh physiotherapy graduates are not prepared to prescribe exercises for people with musculoskeletal pain [16]. The Clinical Pilates exercise method is trademarked by Dance Medicine Australia (DMA) and developed based on the directional preference of an individual with musculoskeletal pain or movement dysfunction [17,18]. A recent systematic review found that general Pilates and the McKenzie method were more efficacious among other exercise interventions [19], and they form the foundation of the DMA Clinical Pilates exercise method. The DMA Clinical Pilates subjective assessment is aligned with physiotherapy practice and has a strong emphasis on the use of a body chart, easing factor of pain, aggravating factor of pain and trauma direction or mechanism of injury [18]. In objective assessments, the movement based classification and treatment (MBCT) approach can complement the conventional physiotherapy objective assessment approach [17]. The assessment primarily evaluates the lower quadrant (lumbopelvic structures) movement control instead of lower back (lumbar) impairments. The MBCT approach uses exercises to assess the movement directional preference of an individual and thus determines the optimal rehabilitative exercise plan for a person. However, previous studies on directional preference are limited to either flexion or extension of the spine for unilateral limb movement, which is either left or right [17]. Clinically, directional preference also exists for trunk lateral flexion and rotation, and these two directions are less explored in the current literature.

This review aims to summarise and rationalise the use of the MBCT approach in clinical practice. Findings from this review can provide structured concepts of the DMA Clinical Pilates exercise method for clinicians or exercise specialists to prescribe rehabilitative exercises for patients with chronic LBP.

## 2. Materials and Methods

A narrative review approach was used because of the scarcity of literature in the investigation of Clinical Pilates for LBP. A literature review was conducted to explore for existing literature that had investigated the Clinical Pilates exercise method for people with non-specific chronic LBP. A search strategy was used: [low back pain (MeSH)] AND [clinical pilates] AND [adult] AND [non-specific] AND [chronic]. This was performed in June 2021 using MEDLINE/PubMed, Cochrane Database of Systematic Reviews/CENTRAL and Cumulative Index to Nursing and Allied Health Literature (CINAHL) for any relevant full-text publications without restriction on the year of publication. The search yielded only two randomised controlled trials (RCT) using the DMA Clinical Pilates method [11,20]. A PRISMA flowchart is shown in Figure 1, and the list of excluded studies yielded from each database can be found in Appendix A. Grey literature search using Google Scholar yielded a single-arm Clinical Pilates study [18]. The included studies were appraised with the Physiotherapy Evidence Database (PEDro) scale for their quality, and their PEDro scores can be found in Appendix A. Studies from the search yield were synthesised for their exercise principles—exercise type, frequency, intensity and duration. Thereafter, the study characteristics are tabulated in Appendix A. This narrative review is structured based on the MBCT approach of the Clinical Pilates exercise method of classifying people with chronic LBP.

## 3. Practical Considerations

### 3.1. Directional Trauma—Clinical History Taking

Directional trauma (or mechanism of injury) of the lower back is categorised by three axes: (1) transverse axis (flexion or extension), (2) anteroposterior axis (lateral flexion to left or right) and (3) longitudinal axis (rotation to left or right), as shown in Figure 2. Exercising in this direction increases pain and movement dysfunction [17], so directional trauma can be used to recognise contraindicated exercises in the early phase of rehabilitation. An individual can have more than one directional trauma, which is a combination of transverse, anteroposterior and/or longitudinal axes.

#### 3.1.1. Transverse Axis Trauma

Movement that involves bending forward (trunk and hip flexion) or backward (trunk and hip extension) can result in injury. The high prevalence of non-specific chronic LBP could result from motor coordination impairment of several muscles during movement [21]. Muscles in the hip or pelvis region potentially influence low back stability with or without painful symptoms.

In standing—for instance, bending forward to pick up a pen on the floor with the knee extended—the movement can be limited by muscles crossing two skeletal joints, commonly muscles that attach to pelvic and tibia bones (hip and knee joints). The shortened rectus femoris muscle at the knee reduces its ability to tension at the hip into flexion due to active insufficiency [22]. Concurrently, the lengthened hamstrings muscle at the knee experiences passive insufficiency that limits its lengthening into the hip flexion [22]. The limitations of the rectus femoris and hamstrings muscles can lead to compensation by the thoracolumbar spine to flex to end range, which can result in the relaxation of the erector spinae muscles at terminal movement [23,24]. The sudden relaxation of the erector spinae muscles leads to abnormal loading on passive structures such as intervertebral disc and vertebrae facet joint [23]. In response to harm, the erector spinae muscles may react with a spasmic response (hyperactivity) to resist trunk flexion movement [24,25]. In this instance, the spasmic muscle may potentially result in painful symptoms. In contrast, individuals with greater spinal flexion and hip flexion flexibility were better protected against erector spinae relaxation because terminal flexion was not reached [26]. However, it is unclear if an individual with greater trunk flexion flexibility is at a lower risk of flexion trauma.

Identifying directional trauma is a challenge. Figure 3 illustrates a football tackle by a defender on the striker. There can be several permutations of directional trauma with other axes, as well as the unilateral trauma. Specific to the transverse axis, the left lower quadrant of the striker is a contact injury into extension, while the right lower quadrant can be a non-contact injury into flexion. In either trauma possibility, the symptomatic side is not always indicative of the side with movement dysfunction. A rehabilitation exercise plan that has exercise movements similar to the directional trauma can lead to adverse outcomes (e.g., increased pain and worsened movement performance) [17]. Thus, using movement-based assessment to identify directional preference potentially hastens specific rehabilitation exercise prescriptions.

#### 3.1.2. Anteroposterior Axis Trauma

Movement trauma into trunk lateral flexion can couple with the transverse axis trauma, which is then classified as a unilateral (one-sided) injury. To illustrate the trauma with axes coupling, a challenge from the right and behind can result in the player receiving the rugby ball to sustain right trunk lateral flexion trauma and trunk extension trauma (Figure 4). There may not be a specific muscle that becomes impaired, but it is likely that injury arises from several muscles that failed to function together after the injury [21].

Specific to trunk lateral flexion, movement can be limited to either side of the trunk after injury. The asymmetrical movements between left and right lateral flexion were found among people with non-specific chronic LBP in past and recent cohort studies [27,28]. The inability of the injured side quadratus lumborum muscle to function normally as an antagonist could result in the asymmetry of trunk lateral movement [29]. In Figure 4, due to the potential left quadratus lumborum muscle strain in the player receiving the ball, it is likely that the trunk range of motion into right lateral flexion will be reduced. However, physiotherapy diagnosis of a single muscle as impairment can be a subset of motor coordination impairment. The variability in electromyography readings of muscle activities were not consistent in explaining trunk lateral flexion asymmetry [29]. Thus, range of motion of the trunk provides valuable information in movement-based assessments in Clinical Pilates.

#### 3.1.3. Longitudinal Axis Trauma

In the longitudinal axis, trunk rotation occurs. Similar to the two aforementioned axes, unilateral rotation trauma can couple with either or both of those axes. Trunk rotation movement in the seated position can result in muscle strain due to fatigue or sudden movement [30]. In the anatomical position, left internal oblique and right external oblique muscles are primary anterior muscles for trunk movement into left rotation. In healthy male adults, the highest activation during left trunk rotation was from the right external obliques, whereas healthy female adults displayed higher activation from the right internal oblique muscle [30]. The higher activation of right internal oblique muscle (antagonist) found in healthy female adults is unexpected, which possibly indicates latent motor coordination impairment that can then lead to future injury incidence. A similar trunk rotation injury affecting the antagonistic internal oblique muscle was found in a tennis player [31], but abdominal muscle injury is considered rare among the athletic population [32].

In contrast to the anatomical position, left trunk rotation in the seated position did not show primary involvement of the left internal oblique muscle [30]. One possibility is that in the seated position, the internal oblique muscle is shortened as compared to the anatomical position. Although active insufficiency is commonly used to describe peripheral muscles, the fast dip and rapid fatigue of the left internal oblique muscle in the study showed that trunk muscles attaching to the pelvis can experience a similar phenomenon [30]. In physiotherapy practice, such variances in motor control can result from the maladaptation of muscles to coordinate and perform trunk movements [21]. Thus, even in healthy individuals, the muscle activation patterns to perform trunk movement can vary. So, the MBCT approach is not limited to people with chronic LBP and is potentially useful in identifying healthy individuals who may be at risk of directional trauma.

### 3.2. Directional Preference—Movement Assessment

It is widely known that conventional Pilates exercises aim to retrain motor coordination through strength and endurance conditioning using a variety of movements. In contrast, the Clinical Pilates exercise approach involves directional preference in exercise prescription, which is exercising into pain-free direction or movement that is easy to perform [17]. The improvement in movement control during functional tests could result from improvement in the motor coordination of several muscles [33]. It is difficult to generalise the improvement of a specific muscle via electromyography [34]. As such, non-specific chronic LBP can result from muscle dysfunction from above or below the lower back in complex musculoskeletal cases.

Directional preference is the opposite of directional trauma. Routine medical or physiotherapy assessments of the lumbar spine movements (active range of motion into trunk flexion/extension, lateral flexion and rotation) provide preliminary insights to which movements show better range and result in lesser pain (preference). Clinical Pilates practitioners use the diagnostic bullseye (Figure 5a) to help chart the directional trauma and preference. In Figure 3, if the directional trauma from the tackle is identified as left extension trauma, (in red), then the movement-based assessment finding will be directional preference into right trunk flexion (in green), as shown in Figure 5b. Although symptomatic pain is on the left trunk, motor coordination impairment can be identified on the right lower quadrant, and exercises are prescribed into right lumbopelvic flexion. This provides an example of when conventional rehabilitation exercise approachs may fail to achieve desired clinical outcomes because intervention is typically prescribed for the symptomatic left trunk and lumbopelvic structures. In contrast, a Clinical Pilates practitioner will recognise the movement dysfunction on the right that can remain asymptomatic and prescribe only right lower quadrant exercises in the early stage of rehabilitation. Understandably, the mechanism of injury might be unclear or not be recalled by a patient. Then directional preference can first be identified to help predict the directional trauma, which can save time through the reduction of assessments to perform [35].

### 3.3. Assessment Algorithms

Only one study investigated a potential algorithm in the use of the Clinical Pilates approach for objective assessment in physiotherapy practice [17]. We pooled together in Table 1 a list of common Clinical Pilates exercises used in the movement-based assessments. Functional tests such as single-leg squat and single-leg hop are used to observe for postural instability to identify the suspected problem side [17]. The supine roll-up exercise assesses a person for trunk flexion directional preference. If the heel of the person lifts off while rolling up, the person is unlikely to favour trunk flexion movements [17]. However, their study did not pursue the use of lower limb movement exercise to categorise unilateral flexion directional preference. Instead, the trunk is placed into lateral flexion to perform a roll-up exercise to determine unilateral bias; the side of lateral flexion that demonstrates better movement quality is the directional preference [17]. In clinical practice, the use of the bug leg exercise is also used to observe for the side with better movement quality [36], which is the problem side. We suggest an alternative transverse axis assessment algorithm, as shown in Appendix A. The rationale is that the problem side muscles serve as stabilisers for the lumbopelvic girdle because the convex sacrum is an unstable support. The problem side stabilisers will then result in poor movement control in the contralateral limb (non-problem side). Palpation technique complements the movement assessment through the finger pulp to feel for abdominal muscle contractions and identify the movement side with the least contractions (most stable), which is the problem side. Understandably, this rationale is counter-intuitive compared to the conventional approach, which advocates for training on the side with poor movement control.

If a person fails the supine roll-up exercise, the prone single-leg kick exercise is used to explore extension directional preference. In the neutral position, the side with poorer movement control of the limb is hypothesised as the problem side and confirmed with repeat testing into ipsilateral trunk flexion, which will result in better movement control [17]. This is in contrast with the supine flexion preference test because, in prone, the pelvis is stabilised bilaterally by anterior superior iliac spine and anterior thigh shank. The non-problem side muscles will not effectively compensate for the poor movement control of the problem side. That said, it is possible that a person who fails the flexion preference test can also fail the extension preference test. The diagnostic bullseye (Figure 5a) shows an inner circle and an outer circle. The outer circle represents full-range movement into flexion or extension preference that is explained in the literature [17]. The inner circle represents mid-range movement into flexion or extension, which is a gap in the current literature.

Hip exercises play an integral role in the management of LBP [37]. The side-lying clamshell exercise is a common rehabilitation exercise performed in 30 to 60 degrees of hip flexion in general exercise prescription for the management of LBP [38]. The use of clamshell exercise for chronic LBP relates to the relationship with gluteus medius muscle weakness [39]. A study found 60 degrees hip flexion as the optimal angle to perform the clamshell exercise [40], but there is a lack of consensus from another study [38]. The lack of consensus is unsurprising because, from the Clinical Pilates perspective, it is likely that participants may demonstrate different directional preferences. Movements that involve several muscles such as the clamshell exercise are dependent on the motor coordination of several muscles and can vary between individuals. Although the side-lying clamshell exercise is used to identify trunk lateral bias in the anteroposterior axis, it can potentially bridge the gap of mid-range assessment and an alternative to assess people who have difficulty lying down into supine or prone position. Furthermore, it can help to differentiate complex cases involving bilateral trauma resulting in opposite directional preference of each side. We propose a feasible anteroposterior axis assessment algorithm, as shown in Appendix A.

Currently, there are no studies with regard to trunk rotation directional preference. In Clinical Pilates practice, the identification of trunk rotation directional preference, if present, is assumed to be similar to the identification of trunk flexion directional preference [17]. For those with flexion rotation directional preference, the supine bug roll exercise is used to identify the side that is easier to perform. The mid-range rotation directional preference, on the other hand, uses the crook-lying knee roll (lumbar roll) exercise. For trunk extension directional preference, clinicians aim to identify the side with movement difficulty as the directional preference. In contrast, extension rotation directional preference follows the principle of trunk flexion directional preference because the pelvic structure loses stability during the trunk rotation exercise. In both flexion and mid-range rotation preferences, rolling the knees to one side—for example, to the left—results in contralateral trunk rotation (to the right), which is similar for the attitude rotation exercise in Table 1. In view of the lack of literature and to guide clinicians in longitudinal axis assessments, an algorithm is developed and shown in Appendix A.

In more complex clinical cases, the involvement of the upper quadrant such as neck and scapulothoracic musculature can be a factor of the LBP. Core muscle activity has been found to improve with scapulothoracic and shoulder exercises in standing [41]. The directional trauma and preference for the neck is similar to the lower trunk, which can also be mapped with the diagnostic bullseye (Figure 5a). Anecdotally, in some LBP cases with associated upper quadrant impairments—for example, whiplash injury without neck symptoms but resulting in LBP—positioning the neck into the directional preference of the person (similar to transverse, anteroposterior and/or longitudinal axes in Figure 2) might improve their movement-based assessment exercises in Table 1.

## 4. Recommendations

### 4.1. Exercise Principles

#### 4.1.1. Exercise Type

The Clinical Pilates exercise approach is an individualised intervention whereby exercises are tailored to address trauma specific to the patient [18,20]. Group intervention can be introduced at the later stage of rehabilitation when the patient is able to perform complex exercises [11]. However, existing studies did not explain the different stages of the Clinical Pilates exercise method. According to the Clinical Pilates training, the exercises are prescribed in three stages [36]. Stage 1 exercises are simple (easy to do) and performed in one direction of an axis; Stage 2 exercises are harder than Stage 1 exercises because of additional movement steps (cognitively demanding), but still performed in one direction of an axis; Stage 3 exercises are more challenging and performed in multiple directions and/or axes [36].

Based on the example in Figure 5b, a Stage 1 exercise could be the right single-leg stretch, as shown in Figure 6a(i). The movement is simple, one step, and in the flexion direction of the transverse axis. Stage 1 exercises are foundational Clinical Pilates exercises used to rehabilitate a person to improve the performance of daily functional activities such as walking and stairs climbing. In increasing the difficulty of a Stage 1 exercise, more steps are added to the exercise while maintaining the single direction of movement, which is shown in Figure 6a. The additional steps challenge the musculature of the body to improve motor coordination. Stage 2 exercises are useful for people who participate in recreational activities or sports, including the football striker example in Figure 3.

The final stage of exercise progression, Stage 3, involves multi-directional movements that can include movement into the trauma direction. This stage is similar to the practice of conventional Pilates whereby exercises are not categorised by movement direction and performed in all directional axes. Figure 6b shows an example of exercise progression, which rehabilitates the person to function well in previously provocative movements (into trunk extension). Although this stage of exercise is critical for competitive sports involving multiple axes of trunk movements, Stage 3 exercises do not form the bulk of the rehabilitative exercise prescription plan. This is because movement into the traumatic direction can result in adverse outcomes such as reduction in physical performance [17].

#### 4.1.2. Exercise Frequency

The three studies investigating the Clinical Pilates approach used a 6-week intervention protocol and were conducted 1 to 2 times a week [11,18,20]. Home exercises complement the low session frequency [11,18], which provides opportunity to empower patients toward self-care. Thus, it is recommended that people with chronic LBP perform prescribed Clinical Pilates exercises at least once a day.

#### 4.1.3. Exercise Intensity

Clinical Pilates exercises are prescribed at moderate intensity based on strength or endurance protocol and are adjusted to minimise pain provocation [18]. Other Clinical Pilates studies did not explicitly state the intensity of their exercise programmes [11,20]. The use of the Borg scale of perceived exertion could help to standardise the description of exercise intensity in studies [42].

#### 4.1.4. Exercise Duration

The Clinical Pilates exercise session caps at 60 min [11,18] but can also be lesser than 30 min [20]. The variation in session duration could be attributed to the choice of Clinical Pilates tool use, mat-based (30-min), equipment-based (60-min) or a combination (60-min) [11,18,20]. The amount of time spent in reassessing the participants of past studies within the intervention duration is unclear. Clinically, about 5 to 10 min is required for reassessment in the physiotherapy session.

### 4.2. Safety Considerations

With the knowledge of directional trauma and preference, it is crucial to minimise exercise or activity movements into the directional trauma. In the cohort study, detrimental effects of exercises performed opposite to the directional preference of the adult participants led to a decline in functional performance [17]. Exercising into the directional trauma may also lead to symptoms such as dizziness, nausea and gut discomfort because of the possible effect on the autonomic nervous system [36]. Hence, the movement-based assessment must be practiced with caution, especially when performing physical assessment into the directional trauma of a person.

## 5. Discussion

This review provided a comprehensive overview of the Clinical Pilates exercise method, particularly the physical examination component leading to exercise prescription. There are limited studies investigating this exercise method and the primary difference from other exercises lies in its exercise principle of exercise type [11,17,18], which is focused on one side to intervene. This is similar to the physiotherapy approach in unilateral intervention, but in the Clinical Pilates exercise method, the symptomatic side is not always the side to intervene [17]. Careful assessment of directional preference can help provide guidance towards accurate exercise matching to produce desired clinical outcomes.

The Clinical Pilates exercise method considers lengthened muscles at risk of motor coordination impairments, which can be identified through directional trauma during physiotherapy history taking [18]. Injuries could arise because the lengthened muscles have exceeded their physiological limits to be protected by stretch reflexes or latency in activation [43]. Furthermore, the loading on the lengthened muscles could lead to autogenic inhibition via Golgi tendon reflex [44]. Subsequently, the dysfunctional or inhibited muscles might lead to motor coordination impairment. The Clinical Pilates exercise method believes that, in shortening these affected muscles, it is easier to perform movements or exercises [17]. In addition, the Clinical Pilates exercises involve movements that cannot be executed by the reflex arc; the exercises require an individual to be focused in performing an exercise in sequences with normal breathing, which thus engages supraspinal control to improve motor coordination [21]. Hence, the primary mechanisms of directional preference in the Clinical Pilates exercise method leverage on the shortening of impaired muscles and the engagement of supraspinal control to improve motor control, thereby improving clinical outcomes in pain and function.

The assessment of anteroposterior and longitudinal axes movement preferences are hypothetical and are based on current findings in assessing the movements about the transverse axis [17]. Hence, this review identified possible directions for future clinical trials to undertake and investigate. A clear assessment algorithm is presented in this review for each axis of movement to provide a structured guide for clinicians and exercise specialists to adopt into clinical practice. For the researchers, the Clinical Pilates exercise method explained in this review provides standardisation for future research investigating its use.

The transition from the assessment of non-specific chronic LBP towards the use of movement-based assessment may save time and improve the outcome of rehabilitation. Whilst current evidence is scarce, the Clinical Pilates exercise method demonstrated similar benefits as general exercises in the longer term [11]. A potential advantage of the Clinical Pilates exercise method is an early functional recovery through matching exercises and physical activities with the directional preference of a person [17]. Maintaining our body wellness is perhaps aligned with the principle of servant leadership, which is to leverage on the strengths of an individual rather than targeting their weaknesses [45]. Hence, rehabilitation exercise prescription in the Clinical Pilates exercise method is both an art and a science.

## 6. Conclusions

The Clinical Pilates exercise method provides an alternative approach towards the management of non-specific chronic LBP. Although identifying the mechanism of injury is common in physiotherapy practice, complementing it with the knowledge of directional trauma and a directional preference assessment algorithm can enhance the planning of physical assessment and the development of exercise intervention. Efficacious, efficient and safe exercises can then be prescribed to an individual to alleviate their symptoms and improve movement performance.

## 7. Patents

Clinical Pilates is trademarked by DMA.

## Figures and Tables

**Figure 1 biology-10-01096-f001:**
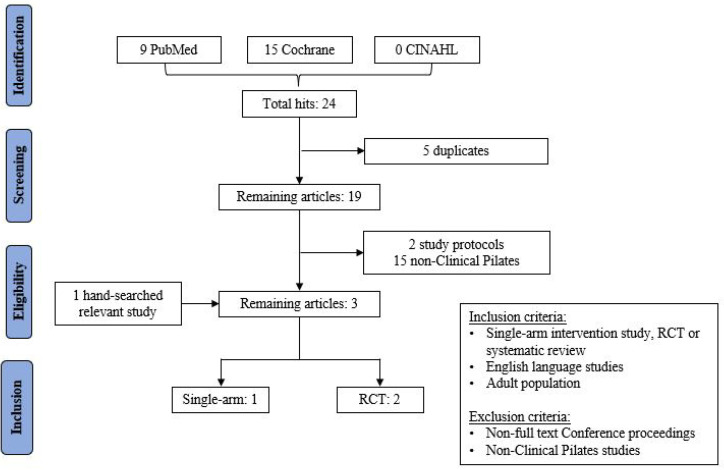
PRISMA flowchart of review search yield.

**Figure 2 biology-10-01096-f002:**
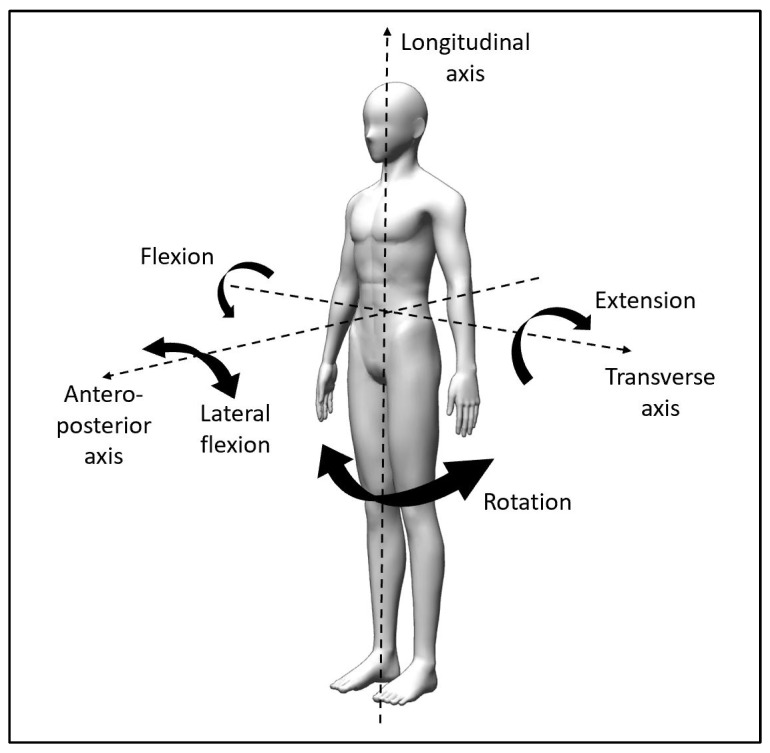
Axes of trunk movements.

**Figure 3 biology-10-01096-f003:**
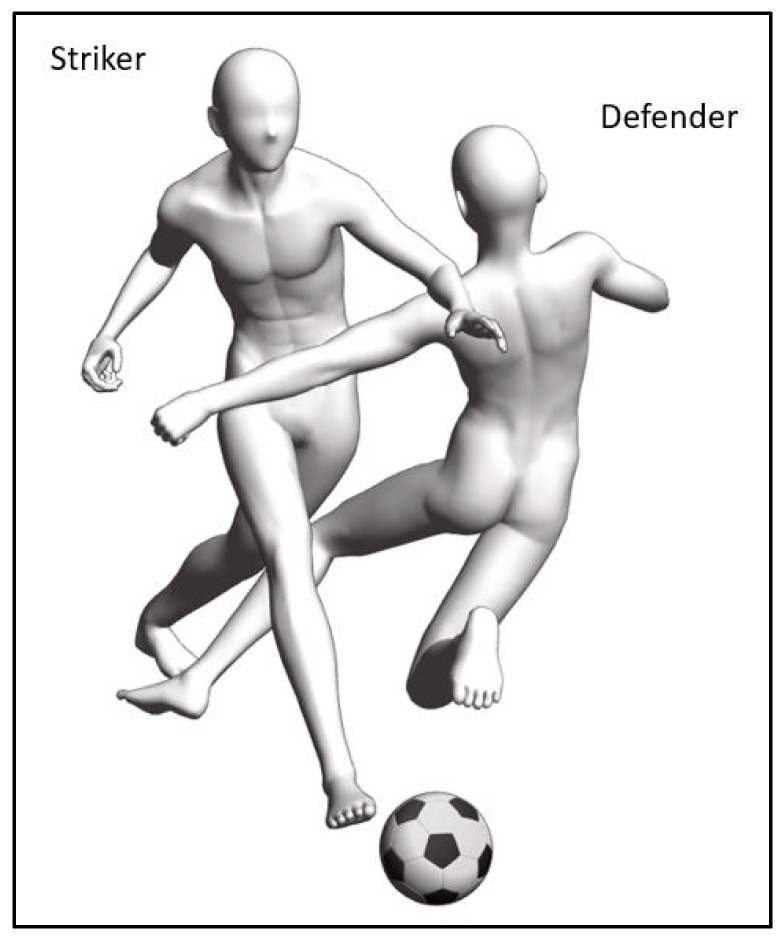
Football tackle from the front and left of a striker.

**Figure 4 biology-10-01096-f004:**
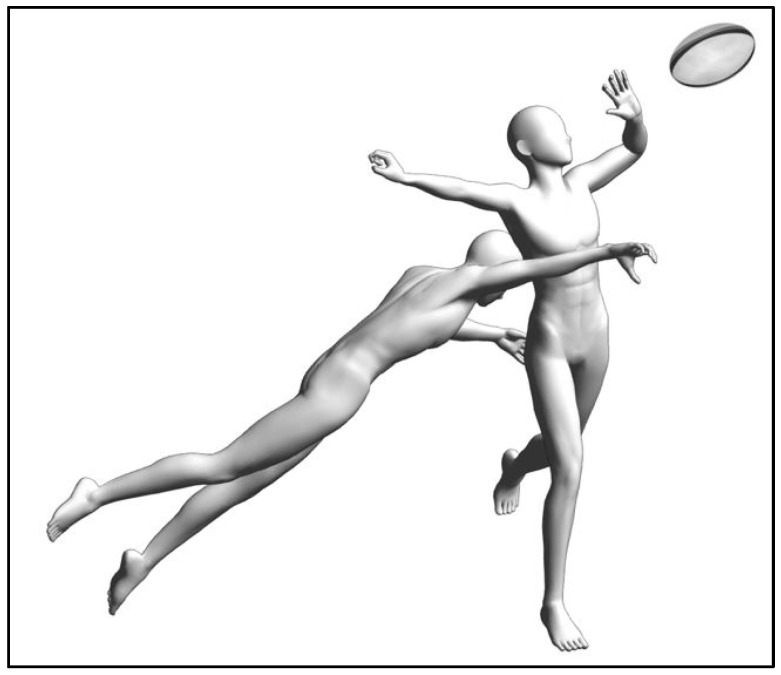
Example of lateral flexion trauma from a rugby challenge.

**Figure 5 biology-10-01096-f005:**
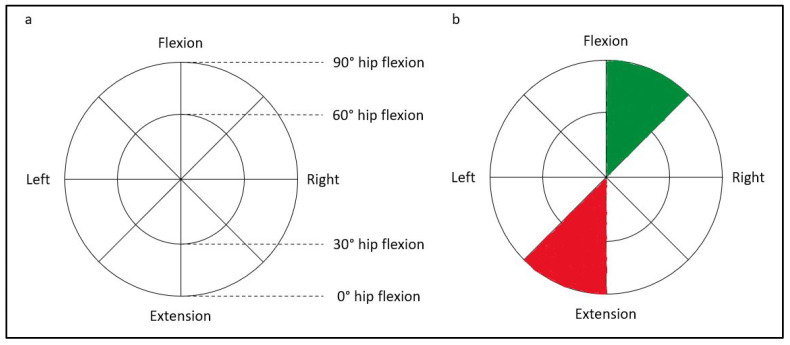
(**a**). The diagnostic bullseye; (**b**). Example of charted directional trauma (red) and preference (green).

**Figure 6 biology-10-01096-f006:**
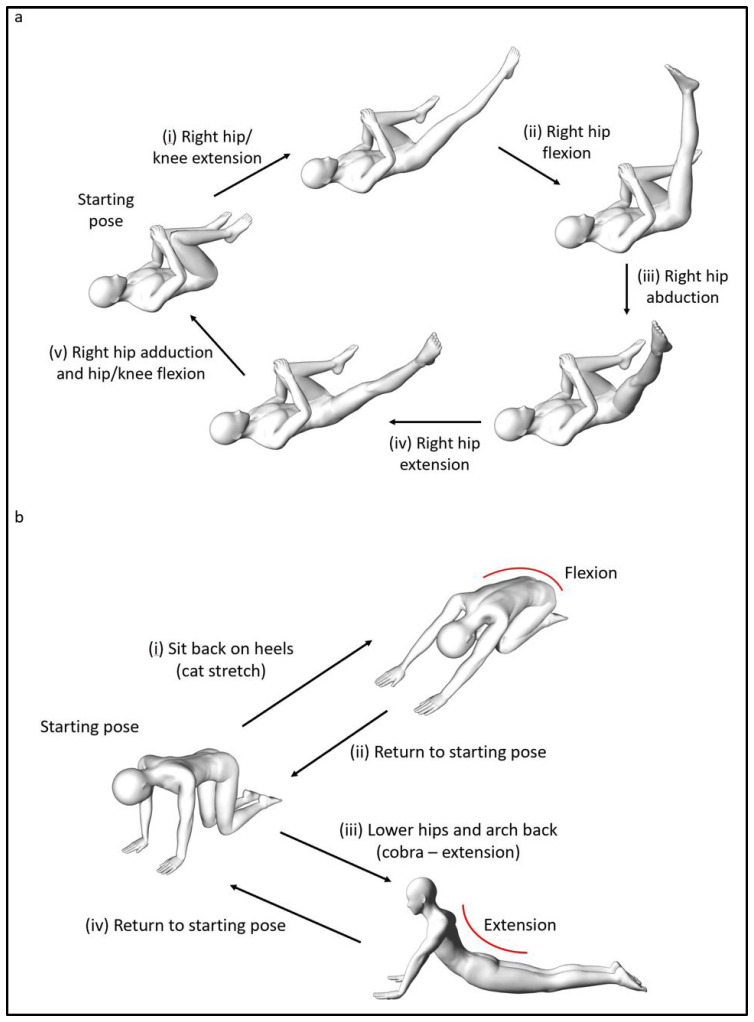
Examples of Clinical Pilates exercises for (**a**) Stage 2 and (**b**) Stage 3.

**Table 1 biology-10-01096-t001:** Summary of Clinical Pilates exercises used in lower quadrant movement-based assessment (excluding upper quadrant variations).

Exercises	Directional Preference Assessed (Axes)	Descriptions
Roll-up 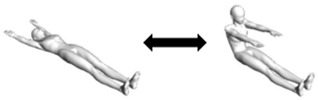 Variations: (i) With one side knee bend; (ii) With trunk lateral flexion to one side; (iii) Combination of (i) and (ii).	Flexion or mid-range flexion (transverse axis). Variations: (i) Unilateral flexion (transverse axis); (ii) Unilateral flexion or mid-range flexion with lateral flexion (transverse and anteroposterior axes); (iii) Unilateral flexion with lateral flexion (transverse and anteroposterior axes).	1. Person lies flat on back with arms stretched out overhead [Starting pose]. 2. Person lifts arms up while flexing head and neck (looking at toes). 3. Person curves upper back (slouch) to lift off from the flat surface with fingers reaching for the toes. 4. Person curves lower back (slouch) to lift off from the flat surface with fingers reaching for the toes. 5. At terminal fingers to toes reach [Ending pose], the person rolls back to the flat surface by reversing the steps described.
Bug leg 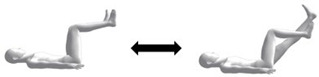 Variation: With trunk lateral flexion to one side.	Unilateral flexion (transverse axis). Variation: Unilateral flexion with lateral flexion (transverse and anteroposterior axes).	1. Person lies flat on back with arms by the side of body and hip/knee flexed to 90 degrees [Starting pose]. 2. Person straightens knee of one leg to between 45 to 60 degrees hip flexion. 3. At terminal knee extension [Ending pose], the person bends hip/knee to the starting pose. Assessor can palpate for abdominal contractions.
Crook-lying leg 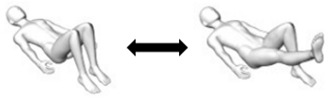 Variation: With trunk lateral flexion to one side.	Unilateral mid-range flexion (transverse axis). Variation: Unilateral mid-range flexion with lateral flexion (transverse and anteroposterior axes).	1. Person lies flat on back with arms by the side of body and hip knee/knee bent with feet firmly placed on flat surface [Starting pose]. 2. Person straightens knee of one leg to between 45 to 60 degrees hip flexion. 3. At terminal knee extension [Ending pose], the person bends hip/knee to the starting pose. Assessor can palpate for abdominal contractions.
Prone single-leg kick 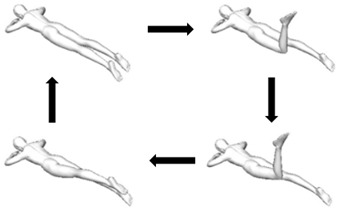 Variation: With trunk lateral flexion to one side.	Unilateral extension (transverse axis). Variation: Unilateral extension with lateral flexion (transverse and anteroposterior axes).	1. Person lies prone on flat surface with back of hands supporting forehead [Starting/Ending pose]. 2. Person bends knee of one leg to 90 degrees. 3. Person lifts bent leg off the flat surface without trunk compensatory movement. 4. Person straightens knee of lifted leg fully. 5. Person lowers leg onto the flat surface gently [Starting/Ending pose]. This test is not applicable in person who is unable to extend hip pass neutral, e.g., with tight hip flexors. Assessor can palpate for muscle contraction near posterior superior iliac spine.
Side-lying clamshell 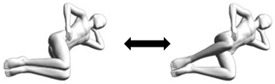 Hip flexion angles: (a) 90 degrees; (b) 60 degrees; (c) 30 degrees; (d) 0 degree. Variations: (i) With trunk lateral flexion (Mermaid); (ii) With trunk rotation; (iii) Combination of (i) and (ii).	Lateral flexion (anteroposterior axis) coupled with: (a) Unilateral flexion (transverse axis);. (b) Unilateral mid-range flexion (transverse axis); (c) Unilateral mid-range extension (transverse axis); (d) Unilateral extension (transverse axis). Variations: (i) As above; (ii) Lateral flexion (anteroposterior axis) with rotation (longitudinal axis) and respective hip flexion angle (transverse axis); (iii) Similar to (ii).	1. Person lies on non-tested side on flat surface with knees bent at 90 degrees and hip flexed at the assessed angle. The palm of one hand supports the head and the other hand on the hip (pelvic crest) [Starting pose]. 2. Person abducts the above knee (hip external rotation as well) to about pelvic level [Ending pose]. 3. Assessor applies break test of the manual muscle testing technique or with handheld dynamometry at the distal thigh of the abducted leg. 4. Repeat the testing at other hip flexion angles to identify the hip flexion angle that produced the peak force (directional preference confirmation) or weakest force (directional trauma identification).
Bug roll 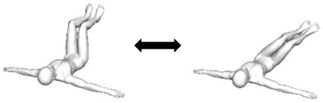 * Variations: (i) Hands cupped at ears (reduce base of support); (ii) Arms crossed on chest (minimal base of support).	Flexion and unilateral rotation (transverse and longitudinal axes).	1. Person lies flat on back with arms stretched out from side of body and hip/knee flexed to 90 degrees [Starting pose]. 2. Roll knees about 30 to 45 degrees to one side [Ending pose]. 3. Return knees to starting pose. Assessor can palpate for abdominal contractions.
Knee/lumbar roll 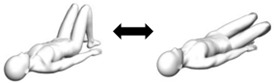 * Variation: Arms crossed on chest (minimal base of support).	Mid-range flexion and unilateral rotation (transverse and longitudinal axes).	1. Person lies flat on back with arms by the side of body and hip knee/knee bent with feet firmly placed on flat surface [Starting pose]. 2. Roll knees about 30 to 45 degrees to one side [Ending pose]. 3. Return knees to starting pose. Assessor can palpate for abdominal contractions.
Prone attitude rotation 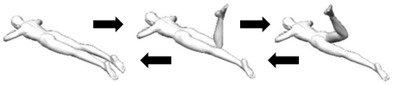 Variation: With arms stretched out from side of body (increase base of support).	Extension and unilateral rotation (transverse and longitudinal axes).	1. Person lies prone on flat surface with back of hands supporting forehead [Starting pose]. 2. Person bends knee of one leg to 90 degrees. 3. Person lifts bent leg off the flat surface and twist to the contralateral side [Ending pose]. 4. Reverse the movement steps to return to the starting pose. Assessor can palpate for muscle contraction near posterior superior iliac spine.

* Increase difficulty with variation if unable to differentiate side of directional preference.

## Data Availability

Not applicable.

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
