# Peer review of "The Theoretical Framework of the Clinical Pilates Exercise Method in Managing Non-Specific Chronic Low Back Pain: A Narrative Review"

_biology, 2021, doi:10.3390/biology10111096_

Round 1

Reviewer 1 Report

Please, see the file attached.

Reviewer 2 Report

I have several comments that need the clarification of the authors.

First, please clarify whether this is a systematic review or a narrative review. If the authors thought theirs to be a systematic review, please format the article according to the PRISMA guideline.

Second, why is the direction of trauma related to the methodology? Please explain. How does the direction of trauma relate to the topic? Please explain.

Third, if this is a systematic review, there are many missing parts. For example, the quality assessment of the included studies is not reported.

Fourth, a table is needed for providing the details of the included studies.

Fifth, the discussion is too short. There is lack of discussion regarding the mechanism of clinical Pilates exercise for clinical improvement.

Round 2

Reviewer 1 Report

Please, see the file attached.

Reviewer 2 Report

The article has been revised well. 

Author Response

Dear reviewer,

Thank you for reviewing our revised manuscript and your positive feedback for our manuscript.